# A Comparison of Approaches to Regional Land-Use Capability Analysis for Agricultural Land-Planning

Tara A. Ippolito [1,2,*], Jeffrey E. Herrick [3], Ekwe L. Dossa [4], Maman Garba [5], Mamadou Ouattara [5], Upendra Singh [6], Zachary P. Stewart [7,8], P. V. Vara Prasad [7], Idrissa A. Oumarou [5] and Jason C. Neff [1,2]

1   The Sustainability Innovation Laboratory at Colorado, University of Colorado at Boulder, Boulder, CO 80309, USA; Jason.C.Neff@colorado.edu
2   The Environmental Studies Program, University of Colorado at Boulder, Boulder, CO 80309, USA
3   Jornada Experimental Range, United States Department of Agriculture—Agricultural Research Service, Las Cruces, NM 88003, USA; jeff.herrick@usda.gov
4   International Fertilizer Development Center, Cotonou 04 BP 673, Benin; EDossa@ifdc.org
5   Institut National de la Recherche Agronomique du Niger (INRAN), Niamey BP 429, Niger; maman_garba@yahoo.fr (M.G.); mouattara@ifdc.org (M.O.); oumarouallelei@yahoo.com (I.A.O.)
6   International Fertilizer Development Center, Muscle Shoals, AL 35661, USA; usingh@ifdc.org
7   Feed the Future Innovation Lab for Collaborative Research on Sustainable Intensification, Kansas State University, Manhattan, KS 66506, USA; zastewart@usaid.gov (Z.P.S.); vara@ksu.edu (P.V.V.P.)
8   Center for Agriculture-Led Growth, United States Agency for International Development, Bureau for Resilience and Food Security, Washington, DC 20004, USA
*   Correspondence: Tara.Ippolito@colorado.edu

**Abstract:** Smallholder agriculture is a major source of income and food for developing nations. With more frequent drought and increasing scarcity of arable land, more accurate land-use planning tools are needed to allocate land resources to support regional agricultural activity. To address this need, we created Land Capability Classification (LCC) system maps using data from two digital soil maps, which were compared with measurements from 1305 field sites in the Dosso region of Niger. Based on these, we developed 250 m gridded maps of LCC values across the region. Across the region, land is severely limited for agricultural use because of low available water-holding capacity (AWC) that limits dry season agricultural potential, especially without irrigation, and requires more frequent irrigation where supplemental water is available. If the AWC limitation is removed in the LCC algorithm (i.e., simulating the use of sufficient irrigation or a much higher and more evenly distributed rainfall), the dominant limitations become less severe and more spatially varied. Finally, we used additional soil fertility data from the field samples to illustrate the value of collecting contemporary data for dynamic soil properties that are critical for crop production, including soil organic carbon, phosphorus and nitrogen.

**Keywords:** land capability classification; drought; land degradation; vulnerability; agriculture

## 1. Introduction

Worldwide, there are over 800 million people who are chronically undernourished [1]. Africa has the highest prevalence of people who are chronically hungry with undernourishment rates in sub-Saharan Africa (SSA) at 23% [1]. Food insecurity is driven by complex interactions of socioeconomic and environmental factors and is exacerbated in places with low adaptive capacity; conditions that are common in smallholder agricultural settings [2]. In SSA, there are a variety of threats to food security occurring at varying scales and magnitudes of severity that enhance the need for multifaceted interventions [3]. These threats—such as the underproduction of crops, climate change, and land degradation—pose unique challenges to food security. Improved land-use planning tools may be one way to address these challenges and the complexities of the agriculture system.

Heat stress and drought conditions threaten the productivity of crops globally through negative impacts on plant growth, physiology and reproduction [4–6]. Average temperatures in Africa are projected to rise faster than the global average with some of the highest increases in temperature to be experienced by the Sahel region [7]. Changing precipitation can include increases in the prevalence of events such as prolonged droughts or intense rainfall and floods, which can irreparably damage crops [8]. In Africa, only 6% of cultivated land is irrigated (NEPAD, 2013). While 8% of natural disasters globally can be attributed to drought, drought accounts for 25% of natural disasters in Africa [9]. Land degradation further increases the sensitivity of agroecological systems to extreme climate events [10].

It is estimated that 3.2 billion people worldwide are negatively impacted by the degradation of the land surface, with a disproportionate effect falling on those who already face poverty [11,12]. About 52% of the global agriculture area is affected by land degradation, including soil salinization, acidification, soil crusting and sealing, compaction, organic matter decline, nutrient imbalance, loss of biodiversity and pollution [13]. Roughly 40% of land degradation has occurred in developing countries and these countries are projected to experience 78% of the global dryland expansion and 50% of the population growth by 2100 [14]. In addition, efforts to feed a rapidly growing human population have typically involved agricultural intensification, which can accelerate land degradation [15] if best management practices are not followed. Assessments of the effects of multiple types of soil degradation have shown yield losses due to the effects of degradation [16]. Improving soil fertility in SSA requires farming systems approaches that prioritize addressing barriers across socioeconomic and biophysical aspects [17]. Since prevention of land degradation is preferable to the restoration of degraded land, land management strategies should attempt to prevent further damage [11,18].

Land-use planning and management strategies aid in the effort to protect or restore soil health and soil fertility for food security, economic growth, and national security [18]. In the face of increasing drought impacts, land capability analysis can be used to assess agricultural potential. If coupled to strategic planning processes, these efforts could contribute to the mitigation of climate and degradation-related risks when combined carefully with appropriate economic and social interventions. The spatial variability of soil poses a challenge for reliably modeling and evaluating soil and landscape processes [19], especially at the small scales at which many decisions are made in smallholder farming settings. This lack of reliability may translate into land-use planning strategies that do not properly address the heterogeneity of the landscape. This problem is further exacerbated by an absence of reliable data, which is common in many parts of sub-Saharan Africa [20]. Some land-use planning tools are regionally specific [21] or crop-specific [22] and some, such as the work presented here, focus primarily on the fundamental capacity of the landscape to support agriculture.

A simple, and widely used approach to land-use planning is the Land Capability Classification system (LCC). LCC is a land potential evaluation system that classifies land based on its limitations for agriculture, including both factors affecting both potential productivity and degradation risk. It has been used to identify and implement management interventions to improve agricultural productivity and sustainability [23,24]. The LCC system is oriented to the assessment of soils and physical land properties, such as slope and texture. The LCC framework is a useful initial analysis of land-use potential and is especially well suited to regions with limited prior land-planning activity and/or limited site-level data. It also provides a biophysical basis for subsequent planning work that incorporates economic and social factors. LCC calculations can be modified to fit the landscape, management approach or crop in question [25]. For example, flooding would not be a limiting factor if a farmer was growing flood-resistant rice varieties but would be a limitation if growing maize. This approach was chosen due to its flexibility and soil-oriented nature, which allows for this work to be easily replicated in any location with any crop.

In this study, we create spatial LCC assessments for agriculture in the Dosso region of southwest Niger. We selected this area because it is important for regional food security and is a target area for multinational investments in improved agricultural output. It is also representative of many areas in Africa where both traditional and digital soil map products have been limited by a relatively low density of soil profile descriptions and measurements. We built an LCC assessment using soil data from analyses of 0–20 cm deep soil samples collected at 1305 sites throughout the region. We then compared this field data-based assessment to LCC assessments built using two popular publicly available global soil maps to demonstrate the opportunities and limitations of using different types of soil data. Our approach serves as a method to incorporate improved understanding of the physical constraints on agricultural land use so that land planning, including the designation of lands suitable for cropping, grazing, and conservation, can be improved.

## 2. Materials and Methods

### 2.1. Study Site Description

The Dosso region of Niger (Figure 1) is the southwestern tip of the nation on the border of Benin and Nigeria. The region is 31,000 km$^2$ and has a population of roughly 2 million. Niger is in the Sahel region of Africa, the ecoclimatic and biogeographic zone of transition nestled between the Sahara Desert and the Sudanian Savanna. Over half of the Sahelian region relies directly or indirectly on agriculture for subsistence, and more than 95% of the agriculture in the Sahel is rainfed [26]. Agriculture in Niger is mostly subsistence farming and consists mostly of cereal crops such as sorghum and millet [27]. Bello and Maman [27] note that cereals take up "90% of the cultivated area per year and constitute the staple diet of the majority of the population". The Dosso region has experienced a reduction in annual rainfall while also experiencing an increase in the number of extremely heavy precipitation days in a number of municipalities [28]. The River Niger runs along the border of the Dosso region and Benin and is subject to annual flooding events [28].

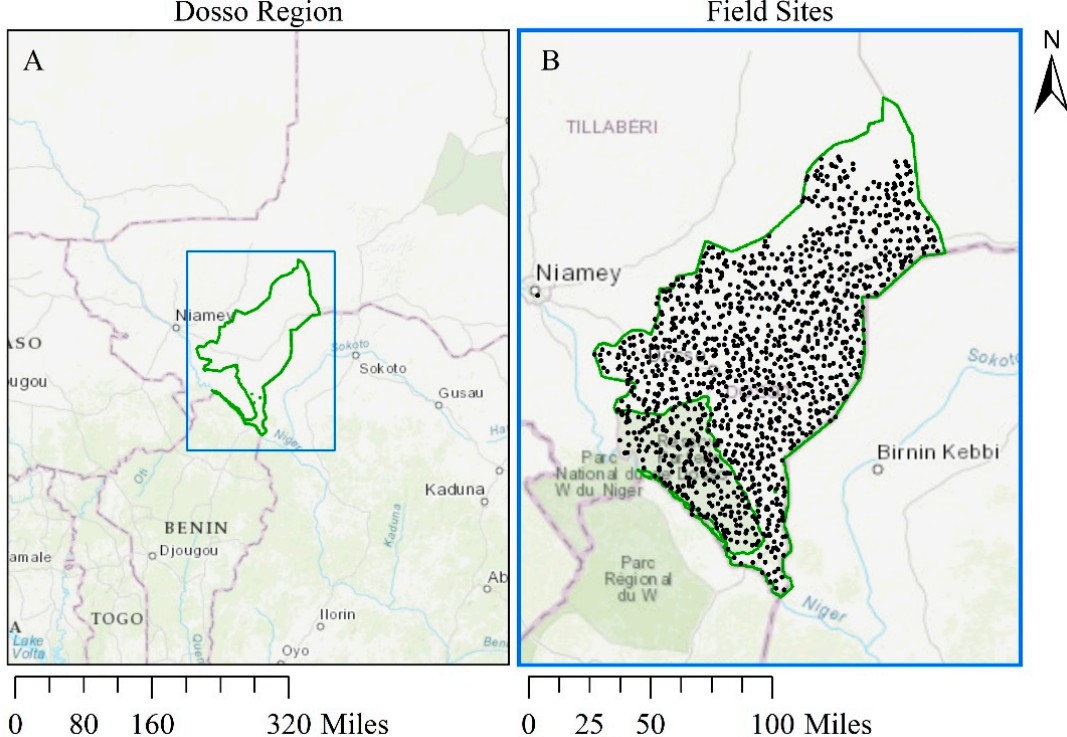

**Figure 1.** (**A**): Boundary of the Dosso region of Niger. Dosso reserve has been removed from Land Capability Classification assessment as it is not considered potential agricultural land, and (**B**): 1305 soil sampling field sites in the Dosso Region.

### 2.2. Field Data

2.2.1. Study Site

The soil fertility survey was conducted in the Dosso region located $13°02'46''$ N and $3°11'50''$ E in Niger. Mean annual rainfall in the region is 635 mm and varies from 350 mm in the Northern part to 800 mm in the Southern part of the region with strong intra and inter-annual variation [29]. Most of the region is arid to semi-arid, and only the southernmost part of the region experiences a higher precipitation regime. The soils in the region include ferruginous soils, hydromorphic soils and newly developed soils from alluvial deposits [30].

2.2.2. Soil Sampling Location

The soil sampling covered the entire Dosso region, except a few locations in the upper North, owing to security threats during the sampling period. A random stratified approach was used for the selection of soil sampling locations. Soil characteristics including clay and soil organic carbon (ISRIC, 250 m resolution) and elevation and slope characteristics from Digital Elevation Model (DEM) content (SRTM, 90 m resolution) were used to stratify sampling across the region; random locations were then generated between the grids to determine the most ideal locations for sampling. Road networks (open street data) were used to determine accessibility, and a buffer of 3 km was used for sample locations. A total of 1305 sampling points were selected (Figure 1B).

Surface soil was collected by means of a stainless steel auger from the 0–20 cm depth. At each geo-referenced sampling location, a composite soil sample was obtained from 10 soil cores randomly collected in a 10 m-radius around the main sampling point. The samples were air-dried at room temperature, sieved through a 2 mm mesh and stored in paper bags for subsequent chemical analysis.

2.2.3. Soil Property Determination

For the LCC calculations and comparisons with the soil maps, soil texture, soil pH, total nitrogen and total organic carbon were determined by NIR spectroscopy, based on calibration from wet chemistry methods using 100 samples, and the associated $R^2$ values. Phosphate and exchangeable bases were extracted with the Mehlich 3 method and determined by inductively coupled plasma optical emission spectroscopy [31]. Because coarse fragments were not measured, we used predictions from the SoilGrids map product for the LCC calculations based on the field data.

2.2.4. Slope Data

For all three products, the slope was calculated by using the "Spatial Analyst Tool Surface Slope" in ArcGIS environment with the most recently available Sentinel-2 12.5 m resolution DEM. High-resolution DEM data was resampled from 12.5 m to 250 m resolution via averaging at subsequent steps in analysis to improve processing speeds and to match the resolution of the soil data used.

### 2.3. Land Capability Classification

The LCC framework groups soils based on their limitations to their use for agricultural production (see Table 1 for a breakdown of classes). The LCC system parses land into eight classes, class 1 being most suitable for cropping and class 8 being least suitable for cropping, based on factors that may limit current production as well as the sustainability of future production. It identifies limiting factors that must be managed (see Table 2 for limiting factors) to reduce degradation risk, increase production, or both. These classes can be used to support land-use planning decisions, technology targeting, and can serve as the first step in determining specific crop and crop production system suitability. Furthermore, determining the limitations of soils provides information about potential interventions which could be used to improve soil capability.

**Table 1.** Abbreviated Land Capability Classification Ranking System (USDA).

| Capability Class Codes [†] | Soil Limitations for Agricultural Use |
|---|---|
| LCC class 1 | Most suitable for cropping systems with few limitations to crop growth |
| LCC class 2 | Suitable for agriculture with moderate limitations that may restrict crop selection or require specific management practices |
| LCC class 3 | Severe limitations that will significantly reduce cropping options and/or require extensive conservation practices |
| LCC class 4 | Very severe limitations with fewer cropping options relative to class 3 and/or more extensive conservation practices |
| LCC class 5–8 | Not suitable to crop cultivation |

[†] Class codes are used to represent both irrigated and non-irrigated land capability classes.

**Table 2.** Land capability classification subclass limitations and possible interventions. For more interventions, see www. wocat.net (accessed on 23 January 2021).

| Subclass | | Potential Interventions |
|---|---|---|
| Erosion | | Stone lines; half-moon; grass bands; zai; reduced tillage; agroforestry; pasture; hay; conservation |
| Soil | Depth | Zai; half-moon; tied ridges; possibly deep tillage (only where depth to a non-bedrock root-limiting layer that can be broken and erosion risk is low); agroforestry; pasture; hay; conservation |
| | Salinity | Plant salinity tolerant crops; modify irrigation schedule and amount, ensure adequate drainage. |
| | Surface stoniness | Remove stones or use planting methods that are not limited by surface stones |
| | Soil water storage capacity | Increase organic amendments such as manure and crop residues; use drought-tolerant crops including pasture and hay species; use zai, stone lines, grass bands, tied ridges, contour ridges and half-moon for rainwater capture; install irrigation; keep soil surface covered; reduce planting density |
| | Lime requirement | Add lime; some biochars in some cases; use non-acidifying fertilizers |
| Wetness | Flooding | Communal level dams, use flood-tolerant crops |
| | Water table depth | Conservation; pasture; hay |
| | Permeability | Increase organic amendments such as manure and crop residues; use zai, stone lines, tied ridges, contour ridges, grass bands, and half-moon for slowing surface runoff; some tillage |

In order to calculate LCC for a given spatial unit, an LCC is calculated for each subclass (Table 2) using data available (Table 3). For example, in calculating the LCC of Lime Requirement, we use pH value to determine the severity of the subclass limitation—pH 5.5–7.2 is considered Class 1, pH 4.5–5.5 or pH 7.2–8.4 is Class 3, and pH < 4.5 or pH > 8.5 is Class 4. The thresholds applied here are from Quandt et al. [25], who developed a global system based on a global review of the implementation of the LCC. The class for a given spatial unit is the maximum (e.g., most restricted) of all the subclasses considered and the subclasses equal to this maximum are the "primary limitations". When subclasses have equally-limiting, maximum LCC values (e.g., erosion and lime requirement are both class 4, which is the maximum LCC values), both subclasses are noted as "primary limitations" and are seen as equivalently limiting. Some subclasses, such as soil depth, are direct measurements while other subclasses are derived. One derived subclass is Available Water-holding Capacity (AWC), which is calculated using measured and derived variables (Table 4) as follows [32]:

**Table 3.** LCC calibration—Field Data Source Attributes.

| Input Data Set | Variables Needed for LCC | Source of Variable |
|---|---|---|
| | Soil Depth | Variable unavailable |
| | Surface soil texture | Sand, silt, and clay percentages from 0–20 cm |
| | Salinity | Variable unavailable |
| | Surface Stoniness [§] | Volumetric gravel content of 0–5 cm horizon [‡] from SoilGrids dataset |
| Field Data | Soil water storage capacity | Calculated for 0–20 cm using texture, organic matter, and rock fragment, multiplied by 5 to have 100 cm of soil water storage capacity |
| | Lime requirement | pH value from 0–20 cm |
| | Flooding | Variable unavailable |
| | Water table depth | Variable unavailable |
| | Permeability | Calculated for each horizon using texture, organic matter, and rock fragment, minimum permeability value of all horizons [‡] used |
| Sentinel-2 Digital Elevation Model | | Calculated using ArcGIS |

§ Surface Stoniness was added to field data through spatially joining surface stoniness measurements from SoilGrids. ‡ SoilGrids have attributes for discrete layers (e.g., 60 cm) rather than attributes for horizons. To calculate attributes for horizons, we took weighted averages of discrete layers as is recommended by Hengl 2017 [33].

**Table 4.** Description of symbols used in the below formulas are.

| Value | Symbols | Units |
|---|---|---|
| Organic Matter | OM (=0.5% & slider) | (%v)—as percent |
| Sand | SAND | (%v)—as decimal |
| Clay | CLAY | (%v)—as decimal |
| 1st Wilting point step | WP1 | (%v)—as decimal |
| Wilting point solution | WP | (%v)—as decimal |
| 1st Field Capacity step | FC1 | (%v)—as decimal |
| Field Capacity solution | FC | (%v)—as decimal |
| Layer Available Water Capacity | LAWC | (cm/cm) |
| Volume fraction of gravel "Rock Fragment" | RF | $(\text{g cm}^{-3})$—as decimal |
| Profile Available Water Capacity | AWC | cm |

$$WP1 = -0.024 * (SAND) + 0.487 * (CLAY) + 0.006 * (OM) + 0.005 * (SAND) * (OM) - 0.013 * (CLAY) * (OM) + 0.068 * (SAND) * (CLAY) + 0.031 \tag{1}$$

$$WP = WP1 + (0.14 * WP1 - 0.02) \tag{2}$$

$$FC1 = -0.251 * (SAND) + 0.195 * (CLAY) + 0.011 * (OM) + 0.006 * (SAND) * (OM) - 0.027 * (CLAY) * (OM) + 0.452 * (SAND) * (CLAY) + 0.299 \tag{3}$$

$$FC = FC1 + (1.283 * FC1 * FC1 - 0.374 * FC1 - 0.015) \tag{4}$$

$$LAWC = ( FC - WP ) * (1 - (RF)) \tag{5}$$

Equations are modeled from standard regressions using the National Cooperative Soil Survey database. Layer available water content is unitless (cm/cm) until it is multiplied by depth (cm). Each Layer's depth thickness (cm) is multiplied by the Layer Available Water Capacity (LAWC). Values for each layer thickness are summed to get the profile available water capacity (AWC). All derived variables in the above equations are dependent upon

organic matter, sand, clay, and a volumetric fraction of gravel; thus, these are the only variables for which direct measurements are needed to calculate Soil Water-storage Capacity.

$$AWCprofile = \sum LAWClayer \cdot LAWC \tag{6}$$

For pseudocode showing how all subclasses are split into LCC classes, please see the supporting information.

To create spatial LCC assessments in the Dosso region, an LCC was calculated for each spatial unit (250 m pixel). We first examined the primary limitation(s) (most limiting subclass) to analyze the most pressing challenges in agricultural use. We then removed the most limiting subclass from the overall LCC calculation to simulate the management of the primary limitation. For example, if the primary limitation was a lime requirement, to calculate the secondary limitation(s), we calculate the overall LCC with all subclasses except the lime requirement. The secondary limitation(s) is the next most limiting subclass LCC. Secondary limitations would show the capability of the land if the most severe subclass limitations were managed appropriately (Figure 2).

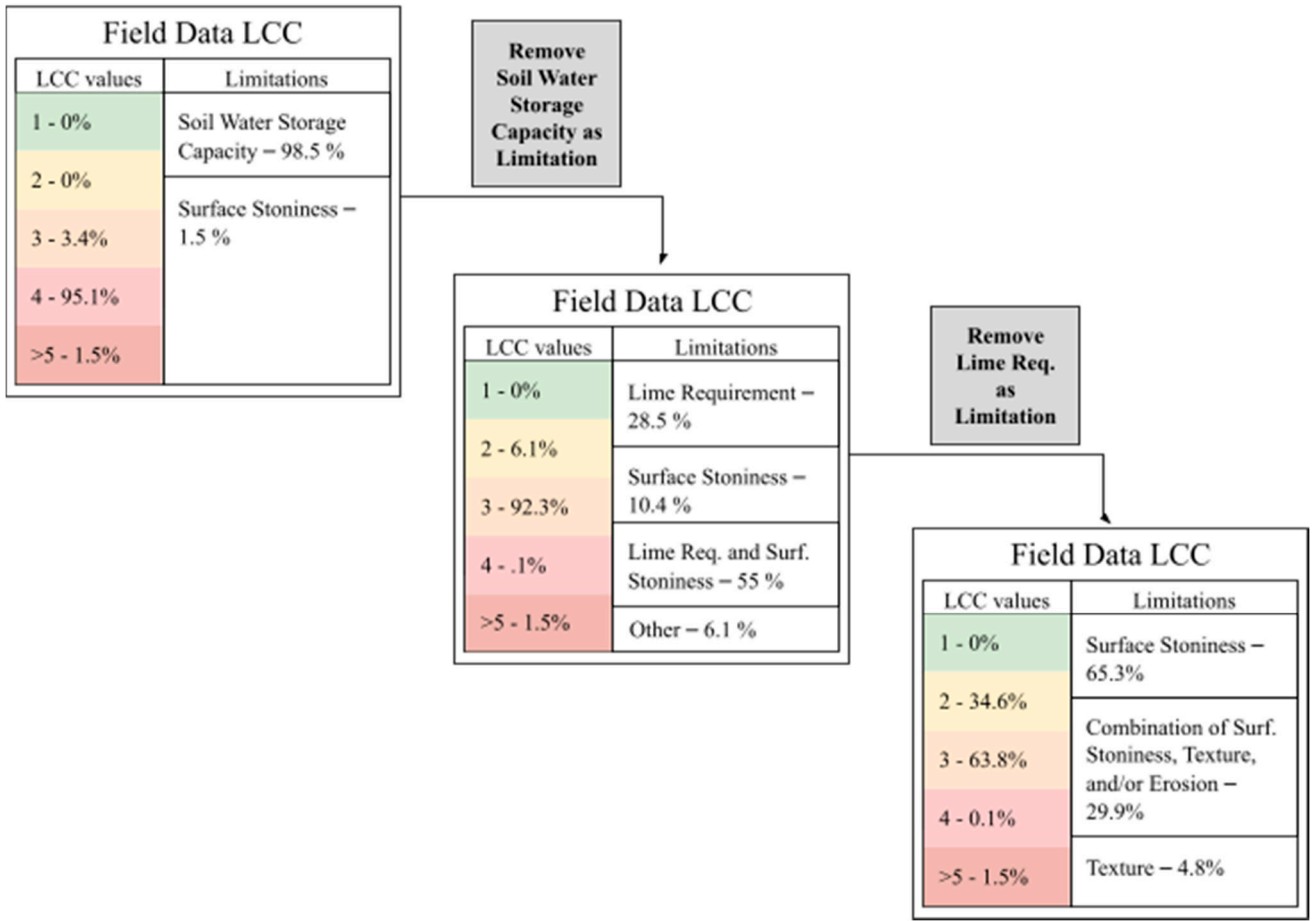

**Figure 2.** Diagram of LCC analysis and the removal of limitations. Each step shows the removal of a primary limitation in each pixel.

The LCC approach typically includes the evaluation of climatic conditions (primarily mean rainfall) as one of the factors influencing agriculture. We did not include this limitation in our LCC analysis for three reasons. First, the majority of this region (with the exception of the southern portion of Dosso) has very little rainfall and crop growth is understood to be constrained by rainfall amounts. Second, rainfall in this region has high interannual variability, making a characterization of average rainfall less valuable for the

prediction of cropping potential than it might be in a more stable climatic region. Lastly, the relative constraint of precipitation depends on the crop and would be best evaluated through a crop suitability analysis. Our approach to LCC (excluding climate) provides a physical system/soil baseline for capability analysis. It also allows for the identification of some key modifiers of site capability, including the water holding capacity of soils.

*2.4. Comparative Digital Soil Datasets*

In SSA, field data is often scarce as it is time-consuming and expensive to collect. In this study, we do a comparative LCC mapping and analysis to show the benefits and limitations of using publicly available digital soil data in lieu of field data. We completed LCC mapping using two commonly used digital soil datasets—Food and Agriculture Organization (FAO) 30 arc-second (~1 km) Harmonized World Soil Database [33] (HWSD) and International Soil Reference and Information Center (ISRIC) 250 m SoilGrids data [34]. HWSD is a traditional global soil map product based on available local and national soil maps. One or more soils is associated with each soil mapping unit and the average percent contribution of each soil is estimated for each unit. We based our analyses only on the dominant soil in each unit. SoilGrids predicts soil attributes using global covariates and common algorithms fitted with local data to develop spatial models of soil properties. Instead of predicting what "soil" will occur at a location, it independently predicts each soil property at each depth for each pixel [35,36]. HWSD and SoilGrids have available attributes that differ from the field data (Table 5) and thus have different LCC algorithm modifications. The same derived slope data from Sentinel-2 DEM was used with both of these datasets as was used in the field data LCC mapping.

**Table 5.** LCC Calibrations—Digital Soil Data Source Attributes.

| Input Data Set | Variables Needed for LCC | Source of Variable |
|---|---|---|
| FAO Harmonized World Soil Database | Soil Depth | Phases—binary indicators based on characteristics that are significant for land management<br>Roots—depth class of an obstacle to roots |
| | Surface soil texture | Sand, silt, and clay percentages for 0–30 cm |
| | Salinity | Electrical conductivity 0–30 cm |
| | Surface Stoniness [§] | Volumetric gravel (particles > 2 mm) content of 0–30 cm |
| | Soil water storage capacity | Calculated for 0–30 cm and 30–100 cm horizons using texture, organic matter, and rock fragment, summed over horizons |
| | Lime requirement | pH value 0–30 cm |
| | Flooding | Phases—binary indicators based on characteristics that are significant for land management |
| | Water table depth | Variable unavailable |
| | Permeability | Calculated for 0–30 cm and 30–100 cm horizons using texture, organic matter, and rock fragment, minimum permeability value of all horizons used |
| Sentinel-2 Digital Elevation Model | | Calculated using ArcGIS |

**Table 5.** *Cont.*

| Input Data Set | Variables Needed for LCC | Source of Variable |
|---|---|---|
| ISRIC SoilGrids | Soil Depth | Depth to Bedrock |
| | Surface soil texture | Sand, silt, and clay percentages of 0–15 cm horizon [†‡] from 0–20 cm |
| | Salinity | Variable unavailable |
| | Surface Stoniness | Volumetric gravel (particles > 2 mm) content of 0–5 cm horizon [‡] |
| | Soil water storage capacity | Calculated for each horizon using texture, organic matter, and rock fragment, summed over horizons [‡] |
| | Lime requirement | pH value 0–30 cm horizon [‡] |
| | Flooding | Variable unavailable |
| | Water table depth | Variable unavailable |
| | Permeability | Calculated using texture, organic matter, and rock fragment [§] |
| Sentinel-2 Digital Elevation Model | | Calculated using ArcGIS |

[§] Surface Stoniness was added to field data through spatially joining surface stoniness measurements from SoilGrids. [†] Since SoilGrids attributes are predicted independently; textures were normalized to 100% as sand, silt and clay percentages often do not add to 100%. [‡] SoilGrids have attributes for discrete layers (e.g., 60 cm) rather than attributes for horizons. To calculate attributes for horizons, we took weighted averages of discrete layers as is recommended by Hengl 2017 [33].

### 2.5. Spatial Analysis

Six distinct data products were created with soil input data from interpolated Field Data, FAO HWSD, and ISRIC SoilGrids. For each dataset, we calculated LCC on a per-pixel basis to generate spatial assessments of primary and secondary LCC values and limitations. All maps are 250 m in resolution. Where input data was not 250 m in resolution, resampling was used to either increase resolution (HWSD resampled from 1 km to 250 m) or aggregated from lower resolution (DEM slope resampled from 12.5 m to 250 m resolution via averaging). Changing the data resolution was necessary for analysis. It had an effect on the analysis outcomes as averaging from higher resolutions to lower resolutions (as in the case of DEM slope resampling) may lead to a relative loss of information. The area of the Dosso Reserve was excluded from LCC maps as it is not considered potential agricultural land.

In order to create a gridded field dataset from the original point data collected, we interpolated each soil attribute using ordinary kriging. We interpolated field data at 250 m resolution in order to match the resolution of the other two LCC assessments. Because the average distance between field sites is 2742 m, the 250 m resolution provided sufficient coverage without multiple points falling within the same pixel. In kriging, first-order trends for sand, silt, and clay values were removed as there is a North/South trend in texture values. Trend removal is a key component of ordinary kriging so that the kriging model is built on the autocorrelation structure of the data; thus, the spatial relationship between sites is understood and used in the prediction of texture values. The North/South trend in texture values is still preserved in resulting predictions. Kriging was implemented within ArcGIS using the Geostatistical Wizard tool.

To calculate LCC for each pixel, we modified the resolution to be uniform across datasets, and layered soil attributes with slope data, so each pixel had all attributes needed for LCC calculation. To layer soil attributes with slope data, all datasets had to be processed in ArcGIS to ensure pixel alignment and to modify resolution as needed. The ArcGIS "Spatial Join" tool was used during data processing in order to layer data. All input datasets were projected onto the same coordinate system in order to ensure there is no geographical displacement between datasets (i.e., all pixel boundaries need to overlap perfectly). Each input dataset was trimmed to the area of analysis—the Dosso region.

*2.6. Fertility Analysis*

While the LCC approach does not specifically address soil fertility (i.e., soil micronutrients and exchangeable bases), these measurements may substantially impact agricultural decisions. To understand the spatial patterns of soil fertility in the region and how they coincide with or diverge from LCC values, we interpolated organic carbon, nitrogen, phosphorus, and calcium point data to 250 m rasters using ordinary kriging in ArcGIS. This approach allows for direct spatial comparison to LCC assessments to see which areas of the map have both high fertility and least restricted LCC values. In order to compare fertility measurements to LCC classes, we discretize each nutrient into 5 nutrient-specific classes (1–5, 1 being the highest-valued) where each class is $1/5$ of the total range of nutrient measurements across the region. Thus, for phosphorus, which has a range of 2.93–7.70 mg P/kg soil, class 1 is 6.74–7.70 mg P/kg soil (high values of nutrients correspond to higher fertility).

## 3. Results

*3.1. Summary of Data Inputs*

One of the main differences between the soil datasets is the topsoil texture (Table 6). The texture is particularly important in this assessment because of its role in water holding capacity and soil drainage conditions. HWSD has far less variability in topsoil texture than SoilGrids or the field data. Across all three datasets, sand content is high with many topsoils classified as sand, loamy sands, or sandy loams. This high prevalence of sand content is reflected in low available water-holding capacity and high permeability of the soils.

**Table 6.** Breakdown of topsoil textures in each dataset.

| Input Soil Dataset | Sand | Loamy Sand | Sandy Loam | Sandy Clay Loam | Loam |
|---|---|---|---|---|---|
| Interpolated Field Data (0–20 cm) | 66.3% | 30.7% | 3.0% | 0% | 0% |
| FAO Harmonized World Soil Database (0–30 cm) | 87.0% | 0% | 5.0% | 0.5% | 7.5% |
| ISRIC SoilGrids (0–15 cm) | 16.5% | 32.3% | 49.3% | 1.0% | 1.0% |

Slopes in the region are mostly flat, with mean slope across the region equal to 2.79% (Maximum slope 29.88%, minimum slope 0%, standard deviation 1.70%). While there are some areas of the region with steeper slopes, the majority of the region has slopes that do not negatively affect land capability classification. When compounded with texture, the slope of the landscape plays a role in erosion risk, which can reduce the capability of land for agricultural usage.

*3.2. Field Data*

For the interpolated field data assessment, the majority of the region (95.1%) is ranked with a LCC value of 4—Very severe limitations—with fewer cropping options and/or requiring extensive conservation practices (Figure 3, Table 7). In addition, in the interpolated field data assessment, a small portion of the region (1.5%) falls into capability Class 5—Not suitable for crop cultivation. The spatial distribution of the LCC analysis shows that the southernmost tip of the Dosso region has lower LCC classes indicating the better potential for agriculture. The most severe LCC values of 5 or above (i.e., unsuitable for agriculture) are mostly located on the southeastern border of the region. The interpolated field data does not have a substantial amount of variability in LCC values as the standard deviation of the LCC values in the region is 0.22.

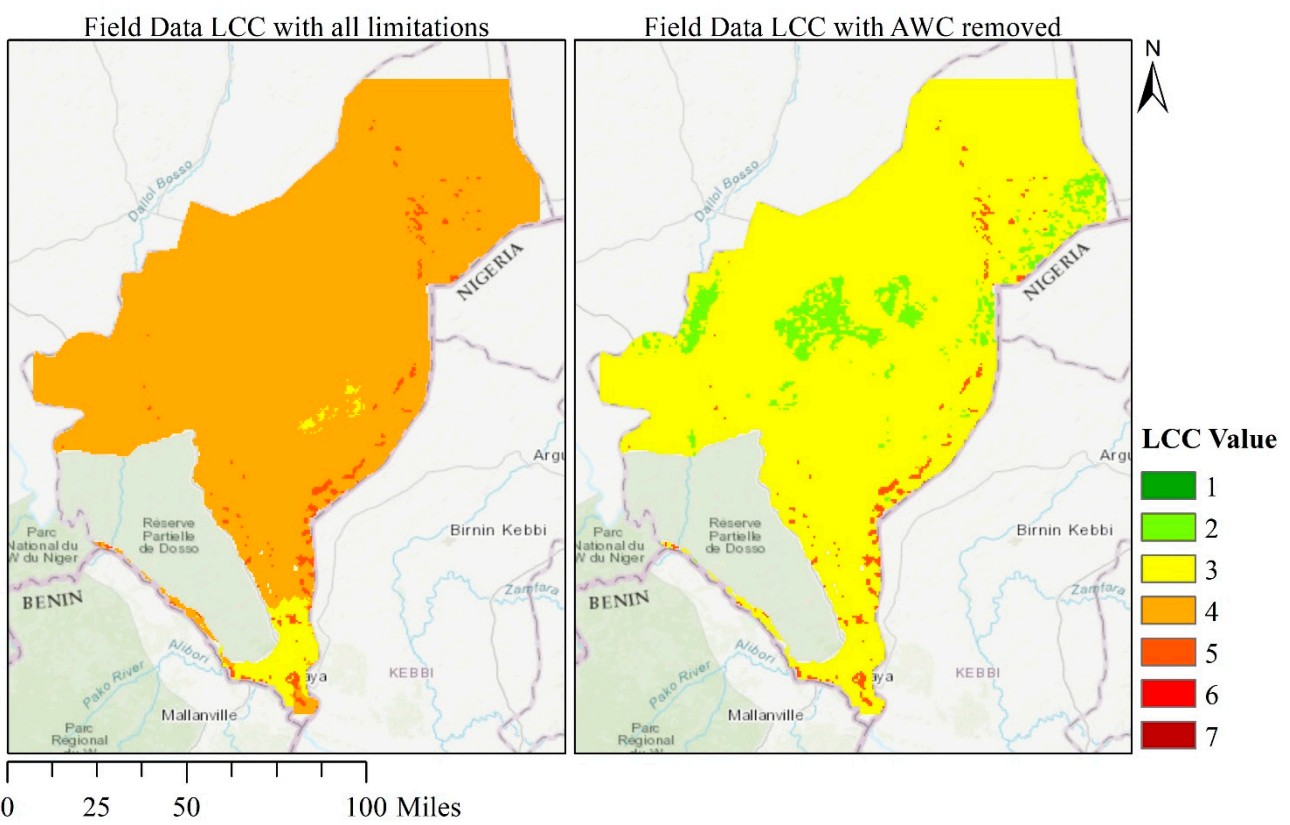

**Figure 3.** LCC mapping of field data with and without AWC as a limitation.

**Table 7.** Land capability classification breakdown with all limitations considered—Field Data.

| LCC results with all limitations considered | | | | | |
|---|---|---|---|---|---|
| **Input soil dataset** | **LCC 1** | **LCC 2** | **LCC 3** | **LCC 4** | **LCC 5–8** |
| Interpolated Field Data (0–20 cm) | 0% | 0% | 3.4% | 95.1% | 1.5% |
| **LCC results with available water holding capacity removed as a limitation** | | | | | |
| **Input soil dataset** | **LCC 1** | **LCC 2** | **LCC 3** | **LCC 4** | **LCC 5–8** |
| Interpolated Field Data (0–20 cm) | 0% | 6.1% | 92.3% | 0.04% | 1.5% |

When the primary limitation, AWC, is removed from the analysis, LCC values improve across the region and across assessments. In the assessments of interpolated field data, the removal of AWC shifts to a majority (92.3%) of the region classified as LCC of 3 instead of LCC of 4. While an LCC of 3 still denotes severe limitations to agriculture, this is a movement in a positive direction for agricultural development. Furthermore, a portion of the region (6.1%) moves into an LCC of 2—moderate restrictions to agriculture.

In the interpolated field data assessment, 98.5% of the region has AWC as a primary limitation and 1.5% with a primary limitation of surface stoniness (recall that there can be multiple primary limitations if multiple subclasses have maximum LCC values) (Figure 4). When we remove the primary limitation (AWC) from the LCC calculation, we investigate the soil suitability if land planners manage for this limitation with irrigation technology and/or timing. We find that in the interpolated field data assessment, the primary limitation shifts to lime requirement (28.5%), surface stoniness (10.4%), or a combination of both (55%), with the remaining 6.1% being a combination of many limitations. When the primary limitation is removed, there is far more spatial heterogeneity in the limitations across the region.

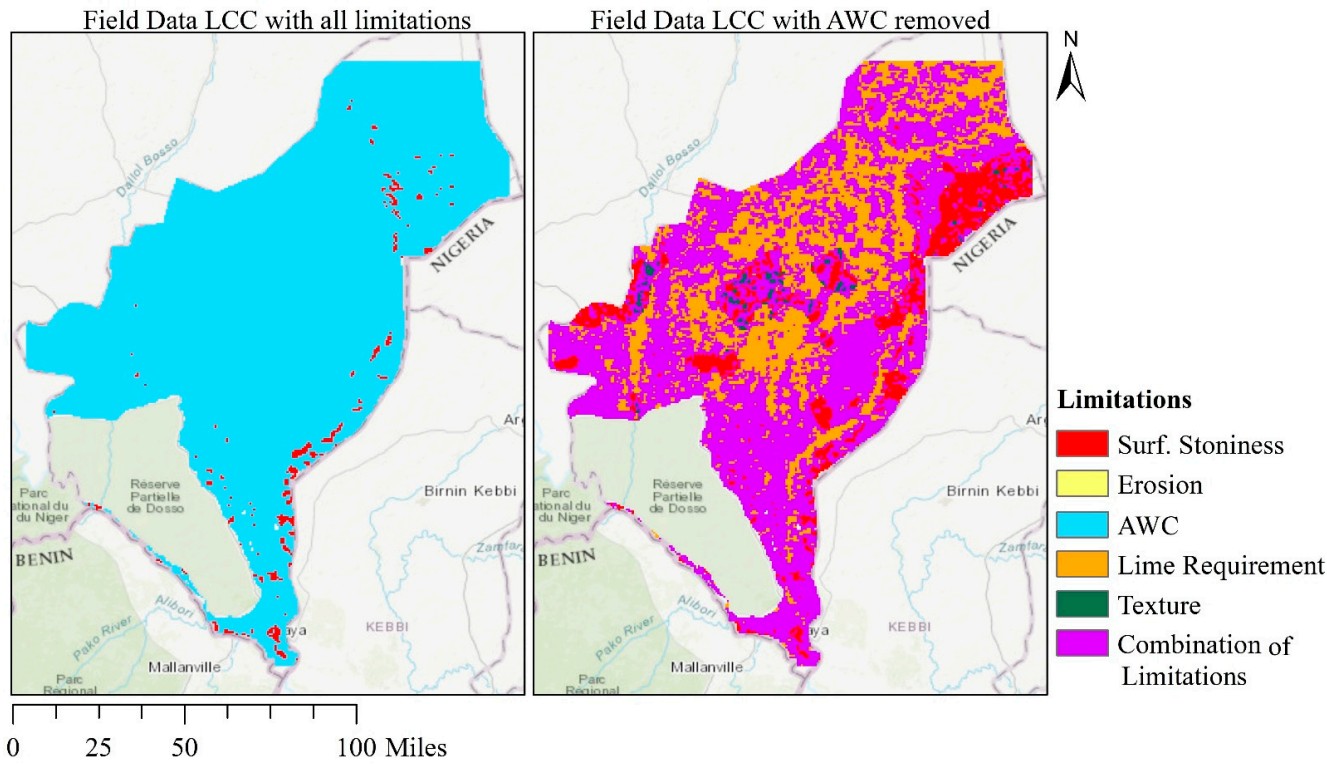

**Figure 4.** Field Data LCC limitations with and without AWC as a limitation.

### 3.3. Digital Soil Datasets

Our investigation found that across all three soil attribute data sets (i.e., HWSD, SoilGrids, and interpolated field data), LCC values are high (meaning poor suitability for crops) in the Dosso region (Table 8). HWSD assesses that a large majority of the region is classified as LCC of 4 (87.5%) and that a small portion (3%) is unsuitable for agriculture. We investigated the LCC values and limitations of subdominant soils (<50% of soil mapping unit) in the HWSD dataset and found only minor differences in severity of LCC classes or limitations. Due to the limited number of soil mapping units within the HWSD dataset, and the fact that we only evaluated the dominant soil in each map unit, the LCC analysis is unable to detect the spatial heterogeneity of LCC severity and limitations. SoilGrids assessments provided the most optimistic evaluation, with 53.9% of the region classified as LCC of 4 and 44.6% classified as LCC of 3. None of the assessments generate LCC values with no (Class 1) or moderate (Class 2) limitations to agriculture.

The spatial distributions of LCC assessments also vary across the three soil data sources (Figure 5). In all of the assessments, the southernmost tip of the Dosso region has lower LCC classes indicating better potential for agriculture. Furthermore, all of the maps have LCC values of 5 or above (i.e., unsuitable for agriculture) on the southeastern border of the region. The differences between spatial distributions of the underlying soil data attributes is mimicked in the spatial distribution of LCC values (Figure 5). SoilGrids has a higher amount of spatial variability (standard deviation 0.52) than the interpolated field data as does HWSD (standard deviation 0.34).

When we remove the primary limitation, AWC, from LCC calculation, LCC values decline across the region and across assessments (Table 8). HWSD and SoilGrids assessments move to the majority of the region classified as an LCC of 3. While HWSD evaluates the region as 97.05%, an LCC value of 3, SoilGrids classifies 66.4% of the region as an LCC of 3 with 32% of the region classified as an LCC of 2.

**Table 8.** Land capability classification breakdown with all limitations considered—digital soil datasets.

| LCC results with all limitations considered | | | | | |
|---|---|---|---|---|---|
| **Input soil dataset** | **LCC 1** | **LCC 2** | **LCC 3** | **LCC 4** | **LCC 5–8** |
| FAO Harmonized World Soil Database | 0% | 0% | 9.5% | 87.5% | 3.0% |
| ISRIC SoilGrids | 0% | 0% | 44.6% | 53.9% | 1.5% |
| **LCC results with available water holding capacity removed as a limitation** | | | | | |
| **Input soil dataset** | **LCC 1** | **LCC 2** | **LCC 3** | **LCC 4** | **LCC 5–8** |
| FAO Harmonized World Soil Database | 0% | 0% | 97.0% | 0.1% | 2.9% |
| ISRIC SoilGrids | 0.06% | 32.0% | 66.4% | 0.04% | 1.5% |

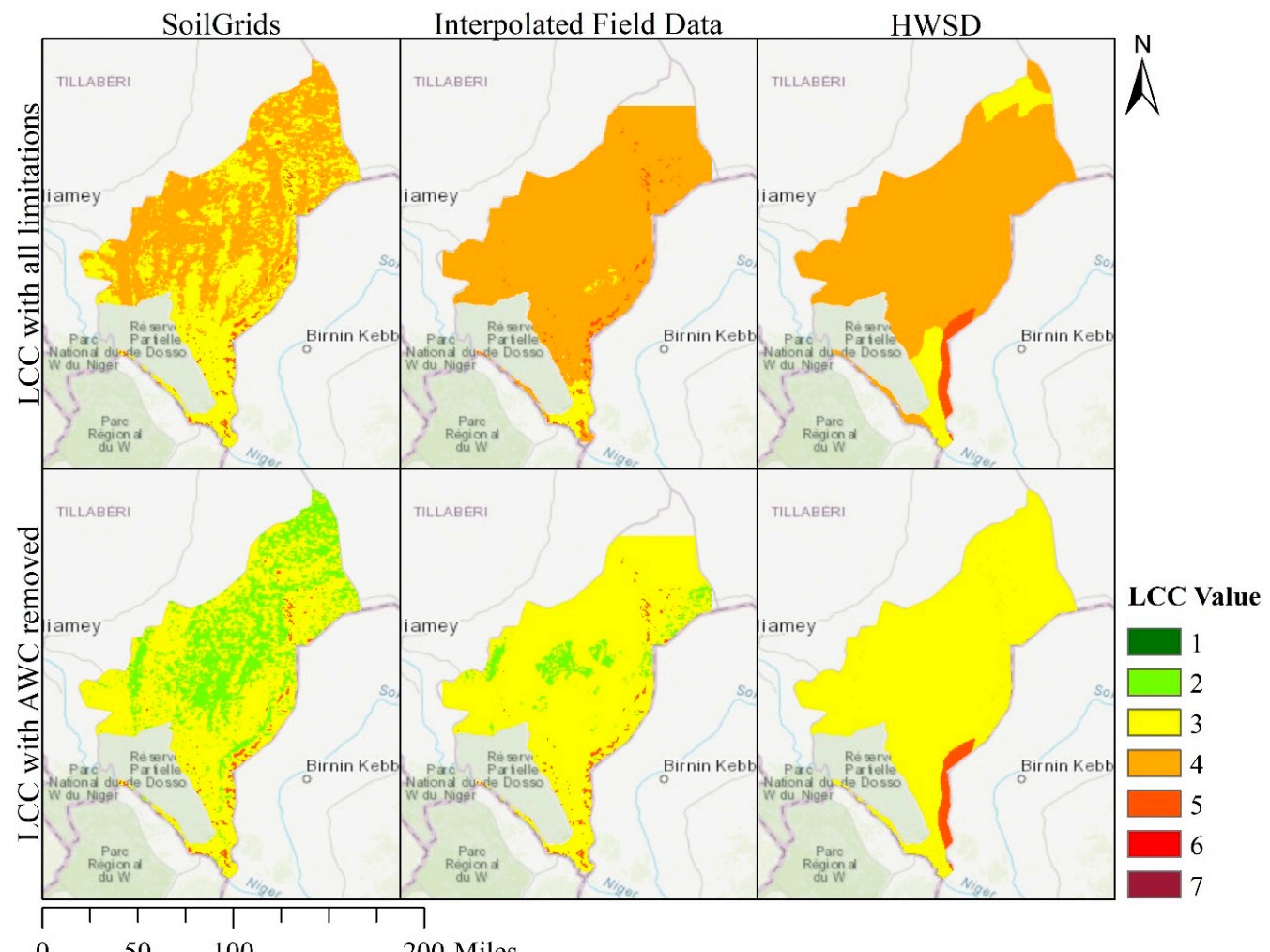

**Figure 5.** Land capability classification (LCC) maps for SoilGrids, field data, and Harmonized World Soil Database (HWSD). LCC ranges from 1 to 8 and is calculated on a per pixel basis.

SoilGrids has an identical breakdown of primary limitations, which is the same as that of the field data since both share the same surface stoniness data. HWSD has slightly higher rates of surface stoniness as a primary limitation (7%) but still assesses the majority of the region (93%) as limited by AWC.

The spatial patterning of limitations is similar across all three assessments with the southeastern tip of the Dosso region showing primary limitation of surface stoniness

(Figure 6). HWSD has less spatially detailed limitations due to the polygon soil mapping unit structure of the underlying soil data. Subdominant soil mapping units did not produce substantial differences in limitation with the exception of one polygon in the southeastern tip where the primary soil unit denotes an impermeable petroferric layer while subdominant units do not. Furthermore, when considering subdominant soils we found little within-map heterogeneity with respect to AWC and as a result, subdominant soils provided little benefit to our analysis.

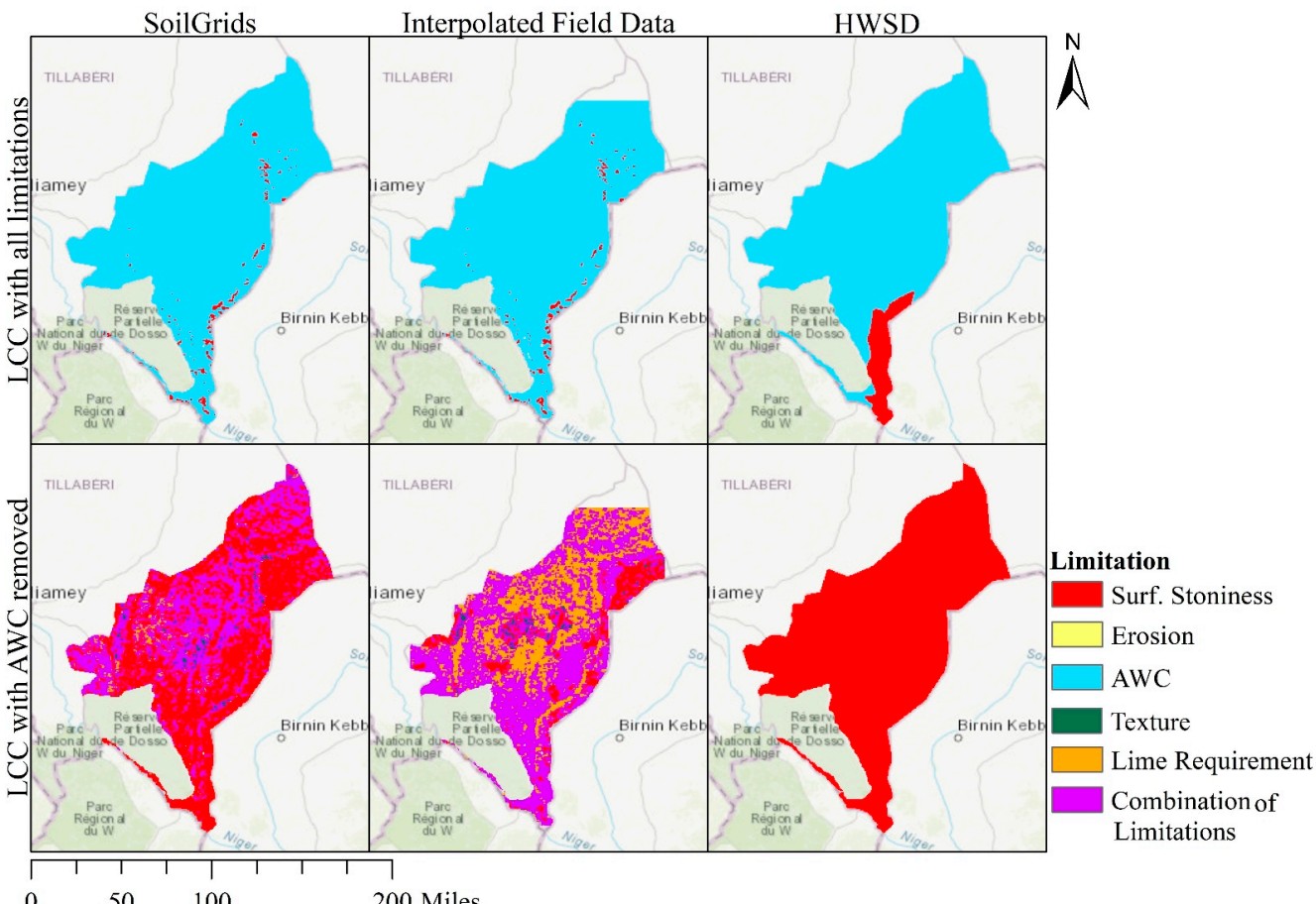

**Figure 6.** Limitation maps for SoilGrids, interpolated field data and Harmonized World Soil Database (HWSD). Note that the field data image uses surface stoniness inputs from SoilGrids resulting in the similar spatial patterning.

In the SoilGrids assessment, we find that 58% of the region is limited by surface stoniness, 28.5% is limited by texture and surface stoniness, and the remaining parts of the region are limited by a combination of subclasses. In HWSD, 100% of the region is limited by surface stoniness. While removal of AWC increases spatial heterogeneity of limitations in the SoilGrids assessment, it reduces spatial heterogeneity in the HWSD assessment.

*3.4. Differences between Datasets*

The primary limitation LCC values are driven by soil water storage capacity which is a function of soil depth, soil texture, volumetric gravel content, and soil organic matter. There are distinct differences in soil water storage capacity between datasets and across the region (Table 9, Figure 7). Since all soil depths were assumed to be the same (1 m), soil texture, organic matter, and volumetric gravel content are the factors which cause discrepancies in soil water storage capacity and thus LCC values between maps. Both HWSD and SoilGrids have more optimistic values of AWC than the field data, with HWSD

values falling completely outside of the field data AWC range (Table 9). While SoilGrids is closer, it is still an overestimate.

**Table 9.** AWC descriptive statistics.

| Input Data Set | AWC Descriptive Statistics (cm/m Soil) | |
|---|---|---|
| FAO Harmonized World Soil Database | Minimum | 8.2 cm |
| | Maximum | 25.4 cm |
| | Mean | 9.4 cm |
| | Standard Deviation | 3.7 cm |
| ISRIC SoilGrids | Minimum | 3.8 cm |
| | Maximum | 11.3 cm |
| | Mean | 6.0 cm |
| | Standard Deviation | 0.9 cm |
| Interpolated Field Data | Minimum | 3.3 cm |
| | Maximum | 7.2 cm |
| | Mean | 4.6 cm |
| | Standard Deviation | 5.0 cm |

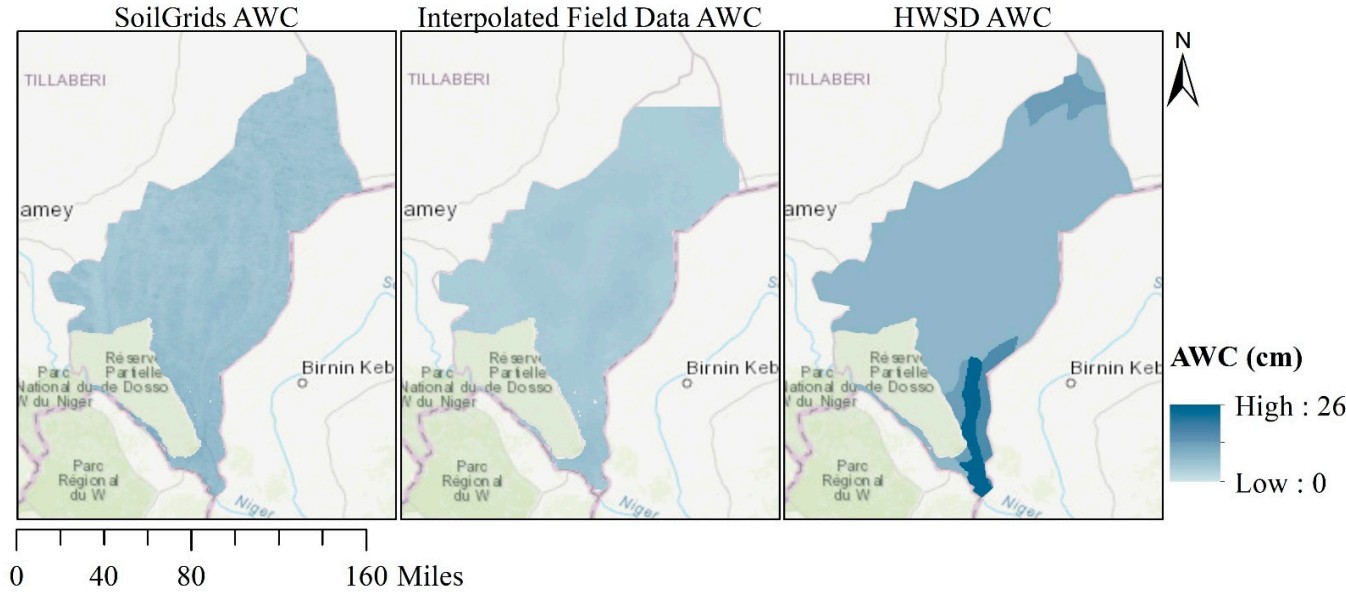

**Figure 7.** Calculated AWC for SoilGrids, field data, and HWSD. AWC is a function of soil texture, rock fragment content, and organic matter. AWC is measured in centimeters and is calculated to 100 cm depth.

Error of the LCC analysis built with digital soil datasets can be measured in two main ways—pixel-level differences in LCC between digital soil LCC maps and field data LCC maps and correlation between digital soil data LCC maps and field data LCC maps. The pixel-level differences are a way of measuring the accuracy in severity of LCC values while the correlation represents the spatial similarities between the maps (i.e., if the whole LCC map is one class less severe than the field data LCC map, every pixel has an error but the spatial structure is maintained). Pixel-level differences show that HWSD is more accurate in assessing the severity of LCC limitations across the region as SoilGrids often underestimates the LCC value (Figure 8). Yet SoilGrids maps are more correlated (pixel-by-pixel correlation) with the field data values as these maps are better at picking up the

spatial heterogeneity while HWSD is blocked out in large polygons (Table 10). Either of these error metrics may be more important based on the goals of the assessment itself.

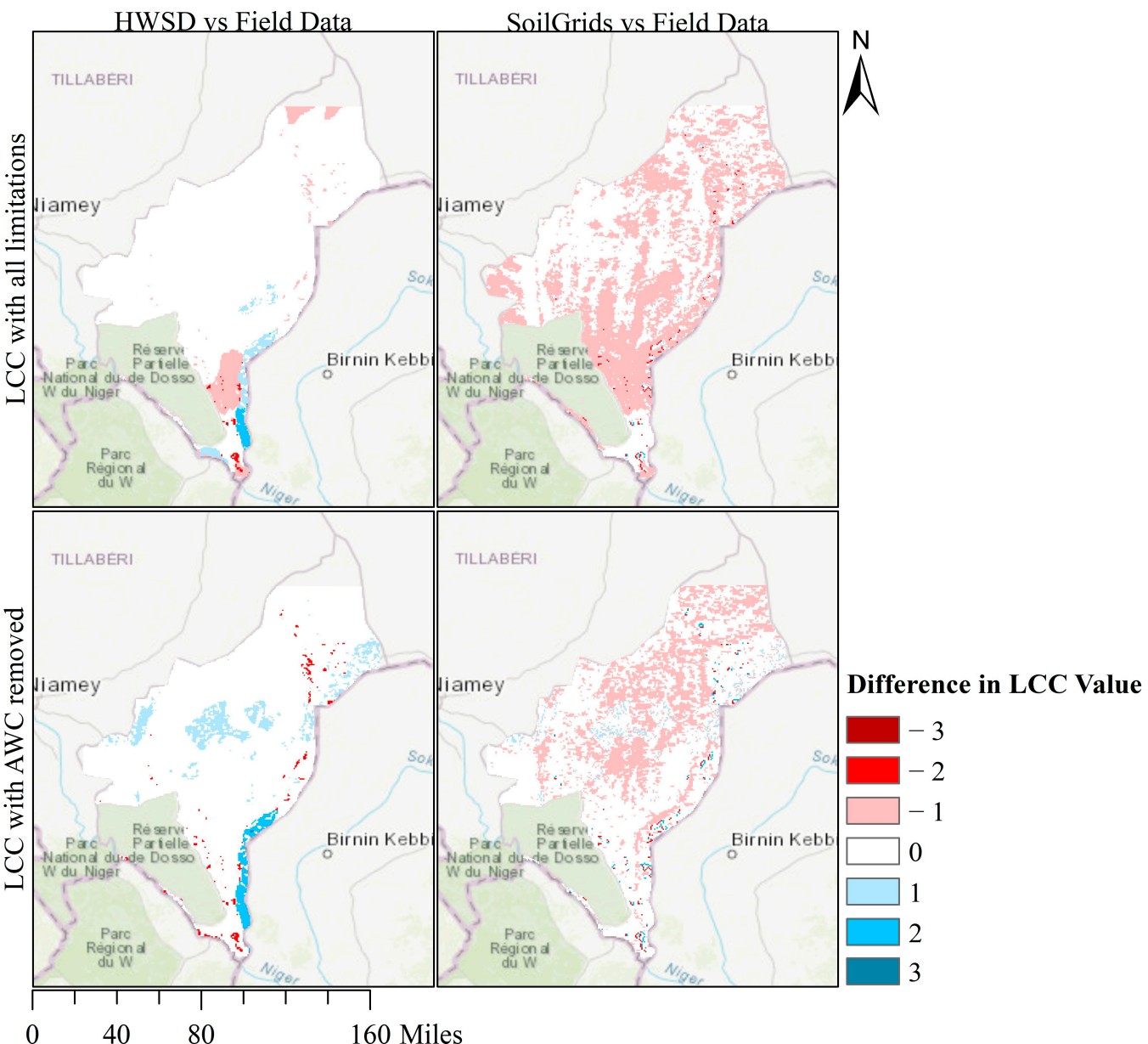

**Figure 8.** Difference between HWSD and field data LCC assessments (HWSD—field) and SoilGrids and field data LCC assessments (SoilGrids—yield) with and without AWC as a limitation. Positive numbers indicate an overestimation of LCC value by the digital products; negative numbers indicate and underestimation of LCC value.

In Niger, soil fertility is generally poor with low mean values of C, N, P and Ca and low variability across the region (Table 11).

**Table 10.** Correlation coefficient between data sets on a pixel-by-pixel basis.

| Data Sets Compared | Correlation Coefficient |
| --- | --- |
| LCC with AWC as a limitation | |
| HWSD LCC and Field Data LCC with AWC | 0.16 |
| SoilGrids LCC and Field Data LCC with AWC | 0.30 |
| HWSD LCC and SoilGrids LCC with AWC | 0.090 |
| LCC without AWC as a limitation | |
| HWSD LCC and Field Data LCC | 0.30 |
| SoilGrids LCC and Field Data LCC | 0.47 |
| HWSD LCC and SoilGrids LCC | 0.25 |

**Table 11.** Interpolated fertility statistics for the Dosso region of Niger.

| Nutrient | Unit of Measurement | Minimum | Maximum | Mean | Standard Deviation | Fertility Class Breakdowns |
| --- | --- | --- | --- | --- | --- | --- |
| Phosphorus | mg P/kg soil | 2.93 | 7.70 | 4.75 | 0.71 | 5: [2.93–3.89]<br>4: [3.89–4.84]<br>3: [4.84–5.79]<br>2: [5.79–6.74]<br>1: [6.74–7.70] |
| Nitrogen | g N/100 g soil | 0.021 | 0.053 | 0.030 | 0.0044 | 5: [0.021–0.027]<br>4: [0.027–0.034]<br>3: [0.034–0.040]<br>2: [0.040–0.046]<br>1: [0.046–0.053] |
| Organic Carbon | g C/100 g soil | 0.23 | 0.74 | 0.35 | 0.070 | 5: [0.23–0.33]<br>4: [0.33–0.43]<br>3: [0.43–0.54]<br>2: [0.54–0.64]<br>1: [0.64–0.74] |
| Calcium | cmol/kg soil | 0.44 | 1.03 | 0.62 | 0.073 | 5: [0.44–0.56]<br>4: [0.56–0.67]<br>3: [0.67–0.79]<br>2: [0.79–0.91]<br>1: [0.91–1.03] |

Fertility maps for the Dosso region based on the interpolated field data show that N, C, and Ca have high values in the southernmost tip of the region (Figure 9). Phosphorus, however, has distinct spatial patterning with highest values in the middle of the region. All of the maps show that the region has low fertility values. With respect to LCC maps, none of the fertility maps closely coincide with interpolated field data LCC maps. The primary LCC analysis which includes AWC as a limitation shows that the southernmost tip of the Dosso region has severe limitations for agriculture (LCC 3) whereas the rest of the region is very severely limited (LCC 4) or unsuitable for agriculture (LCC 5–8). This matches the fertility maps of C, N, and Ca which show highest fertility in the southernmost tip of the region. When AWC is removed as a limitation, high fertility areas in the phosphorus map coincide most closely with low LCC values in the interpolated field data assessment.

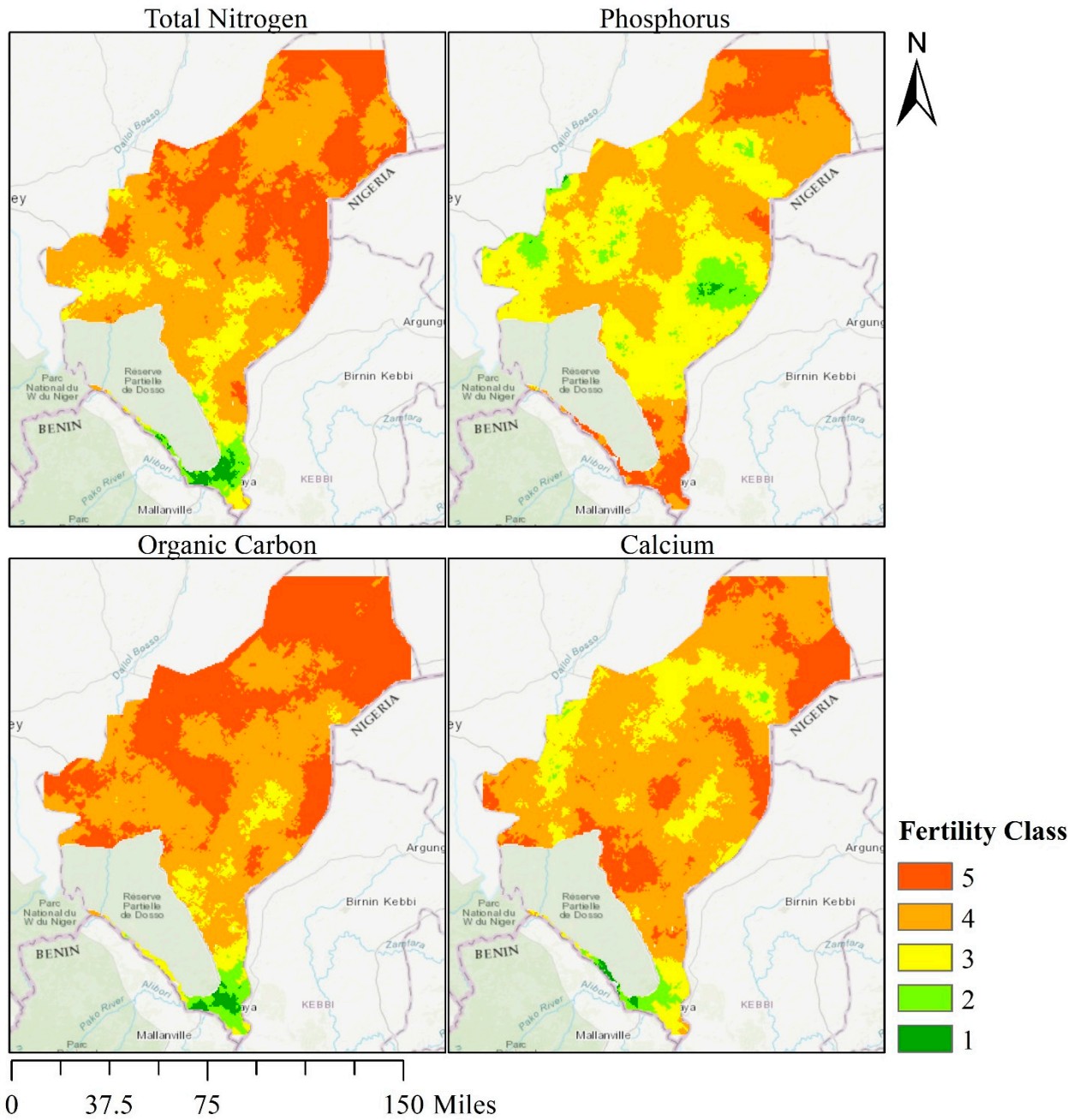

**Figure 9.** Interpolated, discretized fertility maps. Colors represent discretized soil fertility values where fertility class 1 is the highest fertility value and class 5 is the lowest fertility value.

## 4. Discussion

### 4.1. Interpretation for the Dosso Region

Niger's agricultural system has low adaptive capacity in the face of a changing climate due to limited irrigation capacity and low economic development. It serves as a case study for other parts of SSA that face similar constraints. As Niger is projected to reach a population of 65.6 million by 2050 (medium-variant estimate, [37]), there is a growing population of Nigeriens who will need food and employment—both of which can be supported by a strong agricultural system. One of the main solutions presented to stabilize the Sahel region of SSA is to improve agricultural systems and natural resource management [26]. Furthermore, improving agriculture may also help prevent civil unrest as conflict often stems from natural resource scarcity and lack of arable land [38]. The

combined challenges of social conditions and biophysical conditions which limit or stress agricultural production underpin the need for thoughtful management practices. The analysis approach presented in this paper highlights the broad challenges to agricultural development in Niger. However, the constraints on agriculture in Niger are multifaceted and spatially varied and potential interventions and responses may be more effective if these complexities are addressed.

These results illustrate the severe limitations to agriculture in the Dosso region. All 3 LCC assessments show a majority of the region ranking as LCC Class 4. This indicates that limitation-specific intervention strategies will be needed in order to improve the capability of land for agricultural use. In addition, all three approaches indicate that over 90% of land in the Dosso region is primarily limited by AWC and is thus vulnerable to drought. While there is broad agreement that there are severe limitations to agriculture, there is a notable amount of spatial heterogeneity in both the severity and type of limitations when we remove AWC from LCC calculation. This has implications for management as this heterogeneity will affect which intervention strategies are suitable or chosen across the region.

In the Dosso region of Niger, the primary LCC analysis identified AWC, which is a function of soil texture (sand, silt, and clay content), volumetric gravel content, organic matter content and soil depth, as the most prevalent limitation. Since the textures in this region are high in sand content, water infiltration rates are high and retention is relatively low which means that water management is critical to long-term agricultural success. In rainfed agricultural systems, the ability of soil to hold water is crucial in the face of drought events [39] and variation in AWC can be a key factor in determining whether a site is more or less vulnerable to drought conditions. Soils that hold more water may support plant growth for longer periods of time when rainfall is sparse. Possible management responses to the widespread AWC limitation are irrigation, interventions to increase soil organic matter content, use of drought-tolerant or drought-avoiding short-season crop varieties, and use of land for grazing rather than cropping systems.

*4.2. Comparison of the Three Map Products*

The three soil data sources have known limitations that are common to all, as well as individually unique constraints on their use. None of the three data sources includes information on surface stoniness, which LCC defines as the percent surface of the soil surface covered by coarse fragments greater than 25 cm in diameter. Since this measurement was not available in the field data, SoilGrids, or HWSD, we used topsoil volumetric gravel content (percentage of materials in soil which are greater than 2 mm) from SoilGrids and HWSD. In most cases, this will result in an overestimate of surface stoniness due to the inclusion of smaller size classes, but it may also result in an underestimate where coarse fragment content is higher at the surface than in the topsoil, which often occurs in eroded, uncultivated soils.

The interpolated field soil data were further limited by sampling to a depth of just 20 cm. This limitation can have an impact on the results of the LCC as some of the variables, such as AWC, are affected by the characteristics of deeper soils which are unmeasured in the field data. Soils often have textural changes from the surface to deeper layers and these changes can have major impacts on agriculture, particularly when subsurface soils contain fine textures layers that have low infiltration or are in some cases impermeable. For this analysis, we assumed that the subsurface layers were the same texture as the surface 20 cm. For the soil products, we use the mapped properties of deeper layers.

Sampling to just 20 cm also prevented the identification of shallow soils, which are reflected in both a soil depth limitation, and reduced plant-available water holding capacity in LCC. Shallow soils are also missed in many soil maps due to the inherent variability in soil depth in some regions, the fact that shallow soils are often not the dominant soils within a map unit, and because soil mappers are often encouraged to focus their attention on areas dominated by deeper soils with greater potential for agriculture. In the Dosso region, there

are large areas covered by shallow soils that are not fully captured in the field data or the global scale map products. These soils are naturally occupied by shrubs. A limitation of both soil maps is that they are based on soil profile descriptions and measurements (both HWSD and SoilGrids) and field observations (HWSD) that were made decades ago. Due to land pressure, farmers may clear low-quality soils but then later abandon the fields due to a sudden decrease in soil fertility. Such practices leave hardpan and stony areas behind, especially in the northern and central part of the region. Because many of these were degraded relatively recently, their current soil properties may not be reflected in the two soil map products. Similarly, other areas that have been heavily degraded due to land use will not be represented in the HWSD or SoilGrid based products.

There is a disagreement between the three analyses when investigating secondary limitations and these differences could be important especially if additional irrigation is developed in the region. Most notably, LCC maps developed from SoilGrids and HWSD fail to detect the widespread soil pH issues identified from field sampling that would need to be mitigated through the addition of lime. The other key issue that varies between the input datasets is soil texture which has important implications for soil water holding capacity, erosion potential, and other factors. In our analysis of the Dosso region, we find that HWSD texture values are closer to that of the field data than SoilGrids. SoilGrids under-predicts the high levels of sand content and thus presents texture as more favorable to agriculture than the field data suggests. Upon removing AWC as a limitation, SoilGrids continues to underestimate overall LCC constraints while HWSD tends to overestimate the severity of limitations to agriculture. Both HWSD and SoilGrids fail to detect the nuances of secondary limitations to agriculture and particularly the presence of low pH soils in the region that was identified in field sampling. These differences are large enough to generate more favorable LCC ratings for the Dosso region overall.

Both HWSD and SoilGrids lack the detail shown in the field data assessments. This region of Niger may be a uniquely challenging location for the use of both of these products because of limited regional data and issues with spatial resolution. In the case of HWSD, the large spatial scale is a challenge for a regional analysis like this. It is possible that improved consideration of subdominant soils within HWSD could provide useful information which could be incorporated into LCC analysis for added robustness but spatial attribution of dominant and sub-dominant soil properties remains a challenge with this product. While SoilGrids provide higher spatial resolution, there are also large differences between the SoilGrids attributes and field data. In Niger, this may be due to the small number of training points available in the region as inputs to the SoilGrids models. As was noted by Hengl et al. [33], semi-arid and arid areas are often undersampled. It is possible that in other areas of the world, where there are more training samples, SoilGrids would result in estimates that are closer to actual field conditions. Furthermore, the resolution of the digital soil datasets is a limitation in the smallholder setting. In the Dosso region, smallholder farming is the dominant form of agriculture. One 250 m pixel covers roughly 6.25 hectares, which may be larger than a single smallholder farm. For farm-level analysis, higher resolution data or on-farm data collection would be ideal.

While it is ideal, field data is expensive and time-consuming to collect, particularly if it is collected to an appropriate depth for analysis. The limitations in the field data in this study are a common issue across many or most field measurement campaigns as there are economic and logistical constraints to field sampling in all regions and especially in a field setting as challenging as Niger. In virtually all cases, there is rarely sufficient detail in field sampling to create high resolution management strategies. As a result, some combination of field data collection and incorporation of existing geospatial resources may be the most effective near-term regional analysis and mapping strategy.

There are many ways to improve LCC predictions. Most notably, additional, more complete site-level data with a wide variety of attributes and a variety of sampling depths would yield higher confidence in the resulting LCC analysis. In the absence of such data, we suggest that the combination of this type of analysis and targeted field assessment

could yield a viable hybrid approach. The broad-scale analysis here has identified AWC (soil texture) and pH as two key variables. Given this information, high-resolution site-level data in areas of interest could be obtained to identify these limitations. This might include field determination of pH values, and hand textures of soil supported by the use of a mobile application such as LandPKS [18,25]. If the texture analysis was extended to include both surface and subsurface soils, and combined with the spatial data developed here, a reasonably accurate site-scale analysis could be rapidly developed and deployed in conjunction with agricultural interventions such as micro-scale irrigation. At a larger scale, information on water table depth could be used in conjunction with this mapping exercise to localize and prioritize irrigation strategies. LCC analyses can be made more robust through the comparison to fertility data. Measurements of micronutrients and other variables which are not incorporated into the LCC may still be of great value to farmers in planning which crops to use or where to put fertilizers. By adding this analysis, we can visually inspect which areas of the region are most suitable for agriculture as determined by the LCC and if they coincide with areas with high soil fertility.

## 5. Conclusions

The LCC provides a first-step assessment of agricultural potential and identifies the limitations which may impact the usage of land for agriculture at a regional scale. Our results demonstrated that the LCC predictions vary depending on the soil data input source, and that all three sources have significant limitations. The quality of input data will affect the utility of the LCC analysis as a decision-making tool. When all variables of interest are included at sufficient resolution, the LCC can act as a first step in the assessment of agricultural potential. The next steps in a strategic evaluation of agriculture potential would include analysis of access and market issues, including the potential for transport of supplies and goods and the capacity of local government to implement land tenure or agricultural management reforms. For the drier regions of Dosso, this analysis has illustrated that soils in the region have varied potential f to support agriculture with enhanced irrigation from ground or surface water and/or the use of drought resistance varietals or similar agronomic modifications. Furthermore, while a LCC analysis based on remote sensing and soil maps provides a starting point for devising a management strategy and as a first step in land-use planning, local information is necessary for farm-level decisions. Even on small farms, there is often a substantial amount of heterogeneity in soil attributes and management concerns.

Furthermore, when using digital soil datasets, the resolution will not provide information at a scale which is compatible with smallholder systems. In order to fully understand the management considerations, there must be some way of locally assessing capability or input soil attributes. Field sampling at the farm level is one way to do this, but is often not accessible for resource-constrained farmers. Other methods of local assessment that incorporate user-sourced information include using mobile apps such as LandPKS. Understanding the key underlying limitations to land capability is critical in attempting to improve agricultural outcomes and to build resilience to climate change and extreme weather events. Furthermore, understanding the spatial variation in limitations can lead to improved allocation of resources and interventions. Land management plans must be as varied as the landscape itself in order to be efficient and effective. When resources are scarce, targeting high-risk areas with low-cost interventions can maximize outcomes.

**Author Contributions:** Conceptualization, J.C.N., J.E.H. and T.A.I.; methodology, T.A.I., U.S. and I.A.O.; software, T.A.I.; validation, U.S.; formal analysis, T.A.I.; investigation, T.A.I.; resources, U.S. and I.A.O.; data curation, T.A.I., U.S. and I.A.O.; writing—original draft preparation, T.A.I. and J.C.N.; writing—review and editing, T.A.I., J.C.N., J.E.H., E.L.D., M.G., M.O., U.S., Z.P.S., P.V.V.P., I.A.O.; visualization, T.A.I.; supervision, T.A.I.; project administration, J.C.N.; funding acquisition, J.C.N. All authors have read and agreed to the published version of the manuscript.

**Funding:** This study was made possible by the support of the American People provided to the Feed the Future Soil Fertility Technology Adoption (SFT) project funded to the International Fertilizer

Development Center (IFDC) through the United States Agency for International Development (USAID). The contents are the sole responsibility of the authors and do not necessarily reflect the views of USAID or the United States Government. Program activities are funded USAID under cooperative agreement number with USAID (No. AID-BFS-IO-15-00001).

**Data Availability Statement:** Source code is available for download at https://github.com/taraippolito/nigerLCC. (accessed on 23 April 2021). These data were derived from the following resources available in the public domain: https://soilgrids.org/http://www.fao.org/soils-portal/data-hub/soil-maps-and-databases/harmonized-world-soil-database-v12/en/. (accessed on 27 October 2019).

**Conflicts of Interest:** The authors declare no conflict of interest.

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
