# Peer review of "A Comparison of Approaches to Regional Land-Use Capability Analysis for Agricultural Land-Planning"

_land, doi:10.3390/land10050458_

Round 1

Reviewer 1 Report

Proposing solutions to improve crop yields in a depressed area such as the Dosso region is a highly interesting approach.

The work is well planned and presents an adequate development of the methods used. The conclusions respond to the approach and the bibliography is adequate. The manuscript is publishable with minor corretions

To improve the manuscript, I suggeste the authors review the following points

Line 211-217. The information on how to obtain the values to apply them in the formula is not clear

Line 608. The conclusions are very general, more specificity is lacking

Reviewer 2 Report

Dear all,

Thank you for the opportunity to read your paper.

In this work a spatial LCC assessments for agriculture in the Dosso region of southwest Niger was created and then compared with measurements from 1305 field sites. The approach is scientifically useful for understanding the physical constraints on agricultural land use and planning the lands suitable for cropping, grazing, and conservation. The proposed approach and methodology have a relevant scientific value. The analysis is correctly conducted, and the paper is well structured.

Reviewer 3 Report

Dear authors,

You did a nice work in order to compare and spatialize LCCs based on using field data and others datasets, that is really interesting topic, and I have the following comments;  

  • You referred that soil sampling depth was 0-20 cm, is this depth sufficient to assess the agriculture LC?
  • Figure 1. should be improved, the letter A and B must be appeared in the subfigures as you mentioned in the caption, also, the north arrow and scale should be appeared. As you talk about the river Niger in the site description, so it will be good to show it in the map.
  • In line 146, replace climate by elevation and slope characteristic, or modify the text to be clearer.
  • In table1, LCC 5-8 why you don't put it in only one merged class (5) as the main propose is to assess agriculture LCC not to classify the not suitable land.
  • Line 197, Soil Water-Storage Capacity should be Available Water Capacity.
  • In line 237, you said that this region has very low rainfall, while you indicated that the precipitation varied between 350 mm to 800 mm and 90% of the area cultivated by wheat --- Thus, that is a not very low amount of precipitation. On the other hand, this amount of precipitation can not be a limiting factor for cultivation of wheat and also can justify why you don't include the climate factor in your assessment processes.
  • In line 299, oooorganic should be corrected.
  • In table 6 and other tables, please revies the numbers format, if you write a number with one decimal, all the number should appear with the same format, and the same in case of using two decimals.
  • In figure 2, (the subfigure 2 & 3) are exactly the same, why you don’t delete one of them and refer to this below the arrow " e.g., remove lime requirements as limitation"
  • In the table 7 you present 5 classes while in figure 3 you illustrated seven classes, again it will be good to keep only five board classes as the subclass 6&7 where so small even, we cannot see.
  • In figure 4 and figure 6, it is not clear the combination class, (combination of what?)
  • In table 8,--- .003 do you mean 0.03 , also the same comment that I commented before, all the numbers should have the same decimals.
  • In figure 5, why you don’t merge 5,6&7 in only one class, also in sub-figures (interpolated field data) some parts were not spatialized as a limited of point samples in these parts, but you can manage this by ARC MAP to spatialize the whole area, otherwise if you cannot it is not a big issue.
  • In table 9, show the big variations in databases and no one of them are closed to the field data as we are speaking about other study area. Are you have any justification for these interested results?
  • The scales of maps vary between figure and other, I think it will better in further work to fix the scale.
  • In figure 9, You should reefer somewhere to the values of N, P, Ca and OC in each fertility class.
  • It may be good if you can add a small section about the limitations of the current study (as the field data bot include important parameters as depth, salinity, flooding and water table depth).

Finally, you did a nice work with good results and hopefully my comments will be useful for you. Congratulation for this contribution in Land

Reviewer 4 Report

In this study the authors create a spatial Land Capability Classification system (LCC) assessment for agriculture in the Dosso region of southwest Niger. They built an LCC assessment using soil data from analyses of 0-20 cm deep soil samples collected at 1305 sites throughout the region. Then, they compared this field data-based assessment to LCC assessments built using two popular publicly-available global soil maps. The main aim of this work is to demonstrate the opportunities and limitations of using different types of soil data.

This is a very interesting study to improve the land planning around the world, i.e the more suitable land for cropping, grazing, and conservation.

The manuscript is successful in highlighting the main aim and methodological development.

Generally, the paper is well-written.

My main specific comments:

Line 147. Please specify better what do you mean. It is not clear for me.

Line 166-169. Please explain better this paragraph. It is not clear for me how do you work with maps with different resolution

I have not clear how do you can compare field data with soil maps with different resolution. Soil maps are 250 and 90 m resolution while field data have different resolution. You should explain better in this section how you make this comparison and discuss the accuracy of the assessment. (line 245-252).

Line 276. 2.5 Spatial Analysis?

The authors have not clearly explained if there are smallholders in the study area. The authors work with a soil map with a resolution of 250 in this agricultural area. If there are many smallholders, this resolution is not accurate enough in the LCC assessment. This should be discussed in the discussion section of the manuscript.

Round 2

Reviewer 3 Report

Dear authors,

Thanks for improving the manuscript and addressing the reviewers' comments.